# Reduction of Compton Background Noise for X-ray Fluorescence Computed Tomography with Deep Learning

Peng Feng [1,2] , Yan Luo [1] , Ruge Zhao [1], Pan Huang [1] , Yonghui Li [1], Peng He [1,2], Bin Tang [1,*] and Xiansheng Zhao [1]

1   The Key Lab of Optoelectronic Technology and Systems, Ministry of Education, Chongqing University, Chongqing 400044, China; coe-fp@cqu.edu.cn (P.F.); 20152572@cqu.edu.cn (Y.L.); 20162516@cqu.edu.cn (R.Z.); panhuang@cqu.edu.cn (P.H.); 20165953@cqu.edu.cn (Y.L.); penghe@cqu.edu.cn (P.H.); 20152450@cqu.edu.cn (X.Z.)
2   ICT NDT Engineering Research Center, Ministry of Education, Chongqing University, Chongqing 400030, China
*   Correspondence: tangbin@cqut.edu.cn

**Abstract:** For bench-top X-ray fluorescence computed tomography (XFCT), the X-ray tube source will bring extreme Compton background noise, resulting in a low signal-to-noise ratio and low contrast detection limit. In this paper, a noise2noise denoising algorithm based on the UNet deep learning network is proposed. The network can use noise image learning to convert the noise image into a clean image. Two sets of phantoms (high concentration Gd phantom and low concentration Bi phantom) are used for scanning to simulate the imaging process under different noise levels and generate the required data set. Additionally, the data set is generated by Geant4 simulation. In the training process, the L1 loss function is used for its good convergence. The image quality is evaluated according to CNR and pixel profile, which shows that our algorithm is better than BM3D, both visually and quantitatively.

**Keywords:** XFCT; Compton background noise; UNet network; noise2noise model



## 1. Introduction

Combining X-ray computed tomography (X-CT) with X-ray fluorescence analysis (XRF), X-ray fluorescence computed tomography (XFCT) is a novel method to detect early-stage cancer [1–3]. Grodzins L et al. first proposed XFCT [4] in 1983. Then, in 2001, Takeda et al. introduced the principle of XFCT in detail and measured the distribution of iodine in mice with a synchrotron radiation source [5]. In 2010, Cheong proposed the first benchtop XFCT imaging system with an X-ray tube source and verified its feasibility in preclinical applications [6]. Cong et al. successfully reconstructed the concentration of GNPs from 0.2% to 0.5% by using a fan-beam X-ray tube source and parallel single-hole collimation [7]. Deng et al. used a conventional X-ray tube source to detect the distribution of GNPs in mouse kidneys [8].

However, the conventional X-ray tube is bremsstrahlung, causing a lot of Compton background noise, which causes low signal-to-noise-ratio and low detection limits of contrast agents [2]. In order to improve the image quality and detection accuracy of XFCT, Compton background noise removal is an urgent problem to be solved. Normally, for in vivo imaging, the denoising method is to subtract the pre-scan image from the post-scan image. Subtraction operation needs to limit the measured object to the same position and posture, and it must be scanned twice. Repeated scans will expose the patient to excessive X-rays, leading to an increased probability of radiation-induced cancer and metabolic abnormalities. Therefore, it is particularly important to change the denoising method during the imaging process.

In recent years, the deep learning-based noise removal method has been widely used in computer vision and image processing, etc. However, it is difficult to generate clean data sets for XFCT. Seongmoon firstly used deep learning for XFCT images denoising in 2020, which also scanned twice to create datasets [9]. Here, we propose an XFCT image denoising algorithm based on the noise2noise deep learning framework. This method only needs one scan and does not need to produce clean datasets. Both the input and the target are noisy images.

## 2. Materials and Methods

### 2.1. XFCT Theory

XFCT can be seen as a stimulated emission tomography, in which a sample is irradiated with X-rays more energetic than the K-shell energy of the target elements of interest. This will produce fluorescence X-rays isotopically emitted from the sample, and the characteristic X-ray can be externally detected for the image reconstruction [10].

We established a coordinate system *x-y* and a rotating coordinate system *s-t* (Figure 1). The relationship between *s-t* and *x-y* is as follows:

$$s = x \cos \alpha + y \sin \alpha$$
$$t = -x \sin \alpha + y \cos \alpha \tag{1}$$

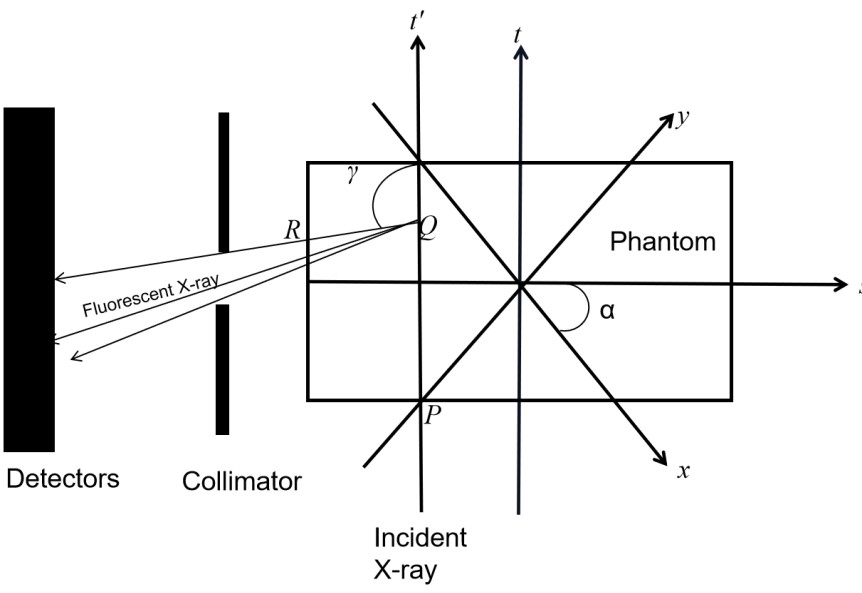

**Figure 1.** Geometry of XFCT.

The process of incident X-rays with an initial intensity of $I_0$ is divided into three steps: the attenuation process of incident X-rays from *P* to *Q*;

$$I_Q = I_0 \exp(-\int_{-\infty}^{Q} \mu^I(s,t) \mathrm{d}t) \tag{2}$$

Fluorescence is excited at point *Q*;

$$I_F = \varphi_Q \rho(s,t) I_Q \tag{3}$$

Fluorescent X-rays reach the detector after attenuation [5]. Then the total flux rate, *I*, of the fluorescent X-ray reaching the detector is:

$$I = I_F \delta(x,y) \int_{\gamma_{\min}}^{\gamma_{\max}} \exp(-\int_{-\infty}^{0} \mu^F(s,t) \mathrm{d}b) \mathrm{d}\gamma \tag{4}$$

Note that $t$ and $s$ variables denote the angle and the translational offset of the incident X-ray, respectively. Where $\varphi_Q$ is the fluorescence yield of contrast agents. Where $\delta(x,y)$ is the angle at point Q viewed by the detector. $\gamma$ is the angle between fluorescent X-ray and $t'$ axis, $\gamma_{min}$ means the minimum angle, $\gamma_{max}$ means the maximum angle. Where $\rho(s,t)$ means the contrast agents' distribution. The reconstruction is defined as an inverse problem in estimating $\rho(s,t)$ from the known linear attenuation of incident X-ray $\mu^I$, and fluorescent X-ray $\mu^F$ and the detected $I$.

In this experiment, the fast multi-pinhole collimated XFCT system is used to simulate scanning the phantom. The system includes an X-ray source, a phantom, two sets of multi-pinhole collimators, and two sets of detectors, as shown in Figure 2a. Two sets of photon-counting detectors are required to obtain projections under double incident photons, thereby reducing the radiation dose [11]. The distance between the X-ray source and the center of the sample is 15 cm (AO), the distance between the collimator and the sample is 5 cm (B1O, B2O), and the distance between the detector and the collimator is also 5 cm (B1C1, B2C2). The detector consists of 55 × 185 detector crystals made of CdTe, the energy resolution is 0.5 keV, the crystal size is 0.3 mm × 0.3 mm, and the center distance of the detection crystal is 0.5 mm. The multi-pinhole collimator is made of Pb with a thickness of 5 mm. There are three pinholes with a radius of 1 mm for a set of multi-pinhole collimators. The pinhole is formed by superimposing two cones at a bottom angle of 55°, as shown in Figure 2b. To avoid overlapping projections on the detectors, the vertical distance between the holes is 1.5 cm.

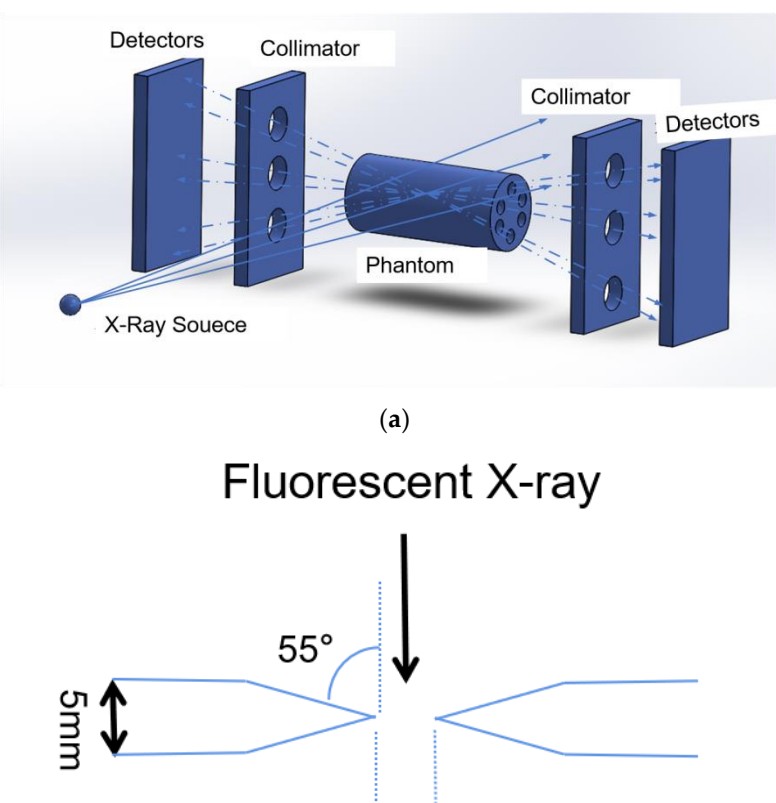

**(a)**

**(b)**

**Figure 2.** Fast multi-pinhole collimated XFCT system, (**a**) Schematic diagram of fast multi-pinhole collimated XFCT system, (**b**) Schematic cross-section of a pinhole.

### 2.2. Noise2noise Model

Noise2noise model was proposed to denoise without a clean target—both the input and the target are noisy images. The network can use the noisy images to learn to convert the noisy input into clean output [12].

When we have clean target $y$ and the noisy input image $x$, we use neural networks to fit the regression model and the loss function is shown as Formula (5).

$$\text{argmin} E\{L(f_\theta(x), y)\} \tag{5}$$

$f_\theta$ is the neural network. If the dependency of input-target pairs $(x,y)$ is removed, Equation (2) is equivalent to:

$$\text{argmin} Ex\{E_y|_x L(f_\theta(x), y)\} \tag{6}$$

In theory, the network minimizes the loss function by solving the point estimation problem separately for each input sample. No matter what particular distribution $y$s are drawn from. Consequently, the optimal network parameters $\theta$ remain unchanged. Therefore, the estimate remains unchanged if we replace the targets with random numbers whose expectations match the targets. Thus, we can replace the clean target with a noisy target whose expectation is equal to the clean target.

### 2.3. Datasets

The data in this experiment is obtained from the Geant4 simulation. Geant4 is a toolkit for the simulation of the passage of particles through matter. It can simulate the process of elementary particles passing through the object, including the collision, excitation, refraction, absorption, etc. Therefore, in this experiment, Geant4 software is used to simulate the process of generating characteristic fluorescence signals to create datasets.

In this study, a phantom with a radius of 2.5 cm was proposed: a small cylinder filled with contrast agents Gadolinium (Gd), a cuboid with length 3 cm, width 1.5 cm, and a height of 5 cm and concentration of 0.2% (Figure 3).

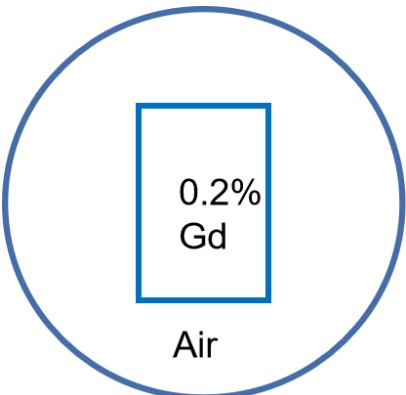

**Figure 3.** Phantom diagram.

Each incident X-ray event is independent, so we set the number of incident photons to 160 billion, and then add the results of randomly selecting 80 billion particles. Consequently, we get the image of 80 billion particles irradiated by the phantom. A total of 1600 images were obtained, which were divided into 1200 to train the dataset and 400 test datasets. The target during training is randomly selected from the train dataset.

### 2.4. Network Architecture

A network similar to UNet is used for denoising. A schematic of the network architecture is shown in Figure 4. Figure 4 illustrates the overview of the denoising network;

the noisy image is used as input to the network, and the output of the network is the denoised image. The network consists of a contracting path (left side) and an expansive path (right side). The corresponding feature maps in the contracting path are copied to the expanding path and then concatenated with the output of the up-sampling operation. The contracting path, like an encoder role, is to extract features of images and then generate deep channel feature maps. It includes five steps: the first step consists of two convolutional 3 × 3 filters followed by a 2 × 2 max pooling function with stride 2 and LeakyReLU. At each down-sampling step (max pooling) we halved the size of feature maps, which will maintain the rotation and translation invariance of the feature. However, steps 2–5 used only one layer of convolution, which could avoid learning redundant features in successive convolution layers. The expanding path, like a decoder, includes five steps. Every step in the expansive path consists of two convolutional 3 × 3 filters, followed by a transposed convolution. After each up-sampling operation, feature map resolution is doubled and the texture information in the image becomes richer. The purpose of the copy operation between two paths is to combine the local and global information of the feature map, and the subsequent 3 × 3 convolution operation better achieves feature fusion. The fusion of shallow edge features and deep high-level semantic features will more comprehensively express the overall information of the image. Because the color information of the input image is simple and consistent, it is not necessary to deepen the number of channels of the feature map. Therefore, we fixed the number of channels of the contracting and expanding paths to 48 and 96, respectively, which not only ensures the noise extraction effect of the network but also significantly reduces the computational complexity of the model. At last, the network outputs a 3-channel denoised image, and the image resolution remains the same as the input.

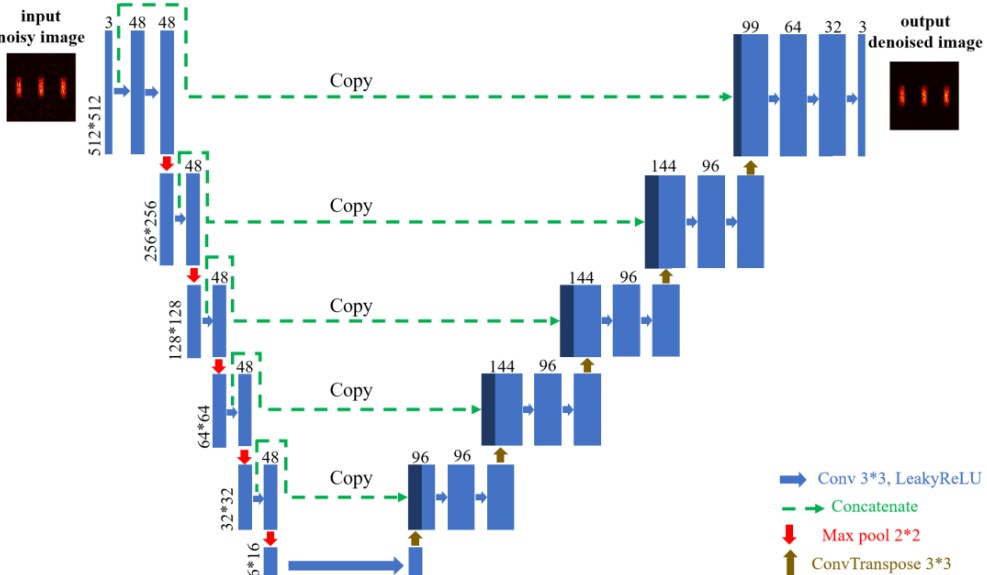

**Figure 4.** The architecture of the network of image denoising. Blue boxes represent multi-channel feature maps, the number of channels is denoted on top of the box. The size (w * h) of the feature map is provided at the left edge of the box. Down and up arrows denote down-sampling and up-sampling operations, respectively.

For the hyper-parameters, we set the learning rate as 0.01, 100 epochs, and 500 batches. We set the iteration number as 600.

In the training stage, we use $L_1$ loss function for its good convergence, which is defined as below:

$$L1(x,y) = \frac{1}{n}\sum_{i=1}^{n}|yi - f(xi)| \tag{7}$$

where $y_i$ is the target value of the model, $f(x_i)$ is the output value.

Since there is rarely research on XFCT image denoising, there is no comparison method, so we compared it with a classical BM3D denoising algorithm. The BM3D algorithm integrates several similar blocks into a three-dimensional matrix by matching them with adjacent image blocks. It performs filtering processing in the three-dimensional space, then inversely transforms and fuses the result to two-dimensional to form a denoised image [13].

## 3. Results

After training in the neural network, we put the test set to verify it. The denoising effect is shown in Figure 5b,c. For the denoised image from the noise2noise model, most of the background noise is removed, the interior of each ROI is uniform, and the boundary between the ROI and the background is clearer than before. Visually, Figure 5b has less background noise than Figure 5c. We used the pixel profile as the subjective evaluation index to observe the grayscale changes of a row or column of pixels. In the pixel profile, the abscissa is the pixel label, the selected pixel is a row, and there are 512 pixels in total, and the ordinate is the pixel value. The image size is 512 × 512, and we select the 256th row to display (this row is marked with a blue line in Figure 5).

It can be seen from Figure 6 that our proposed algorithm can effectively suppress Compton background noise, and a large number of noises outside the region of interest are removed. In the region of interest, after denoising by this algorithm, the gray curve is smoother, the internal pixel distribution is more uniform and closer to the original pixel value. Obviously, the denoising effect of the BM3D algorithm is not satisfactory whether in or out of the region of interest. There are still a large number of background noises and isolated noise points in the graph.

We use the contrast to noise ratio (CNR) [14] as an objective criterion to evaluate reconstructed images, which is defined as follows:

$$CNR = \frac{|\Psi_A - \Psi_B|}{\sigma_{bk}} \tag{8}$$

where $\Psi_A$ is the average value of pixels in ROIs, $\Psi_B$ is the average value of background pixels, and $\sigma_{bk}$ is the standard deviation of the pixel values of the background area.

We divide the images displayed by the three pinholes into three ROI regions, as shown in Figure 7a. We calculate the average value of the three ROI regions of all test set images, and the CNR of the three images is shown in Figure 7b. To sum up, the CNR value of the second ROI is the largest, and the first and third ROI values are symmetrically distributed, which is also in line with the actual situation. After denoising, all CNR of three ROI increases, indicating that the image quality is improved. The CNR value of our algorithm is greater than that of the BM3D algorithm.

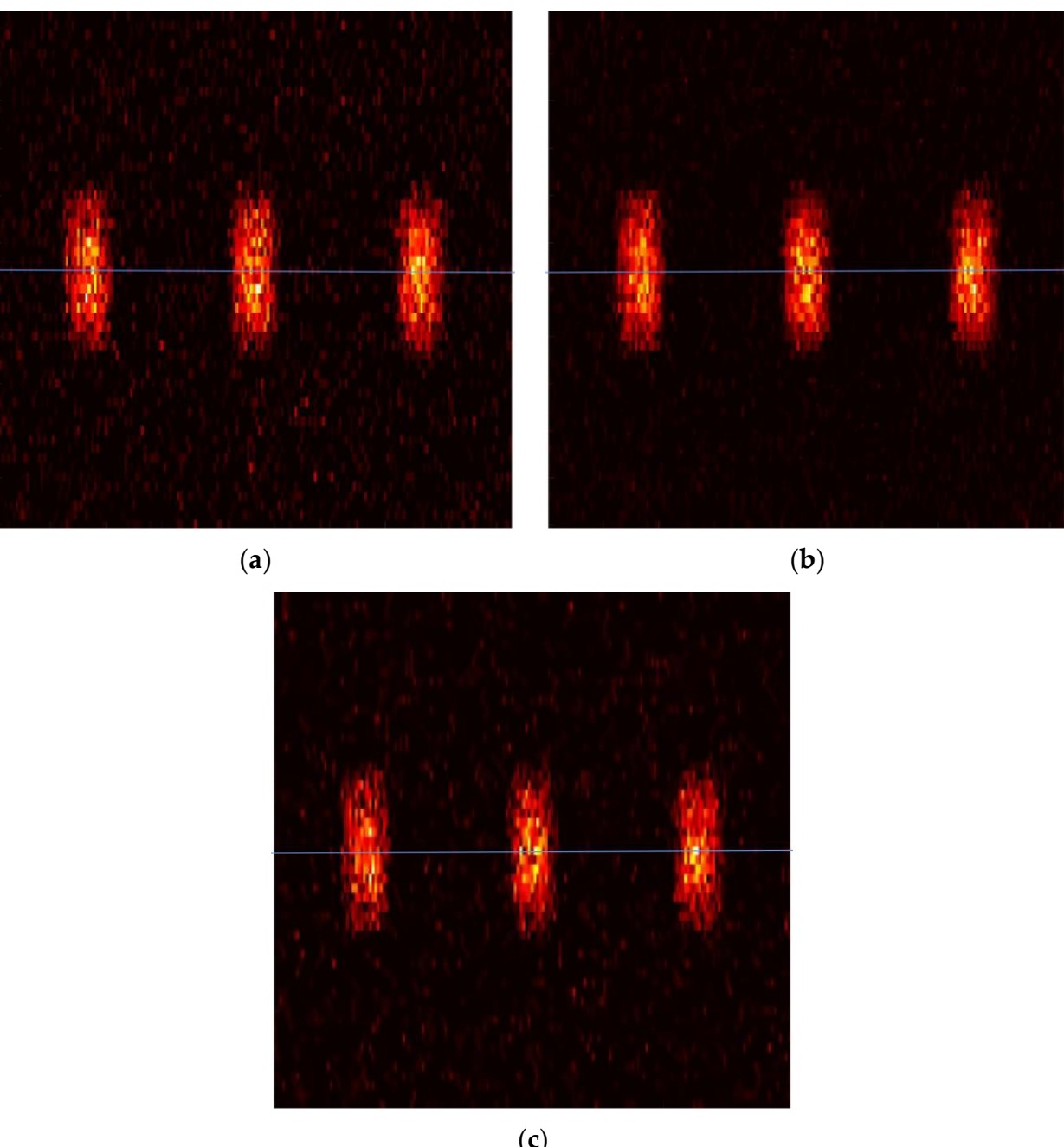

**Figure 5.** Denoising results of Gd phantom image. (**a**) noisy image; (**b**) denoised image (noise2noise); (**c**) denoised image (BM3D).

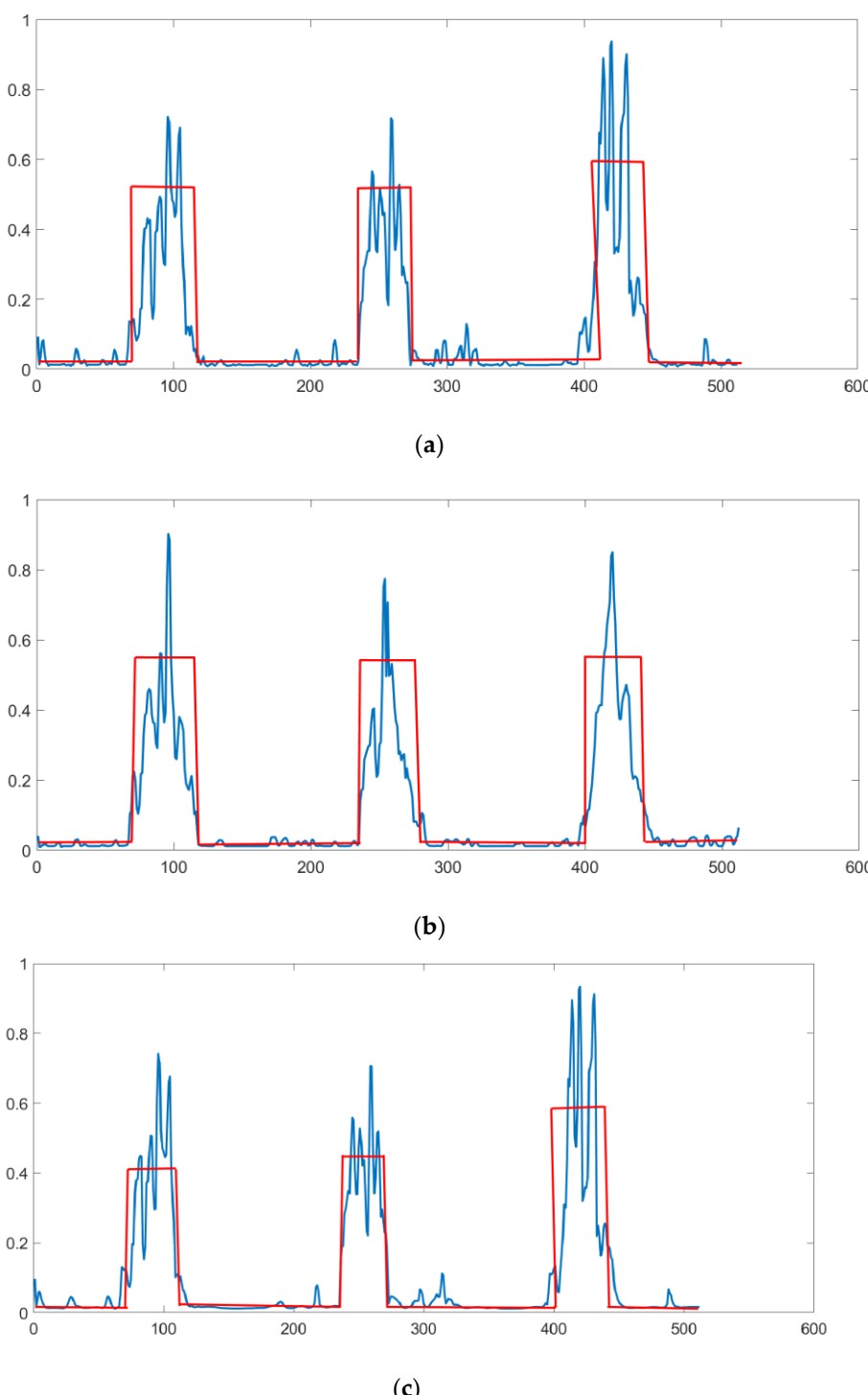

**Figure 6.** Denoising results of Gd phantom image, pixel profile of images (the blue line represents the true pixel value, and the red line represents the standard value) (**a**) pixel profile of noisy image; (**b**) pixel profile of denoised image; (**c**) pixel profile of denoised image (BM3D).

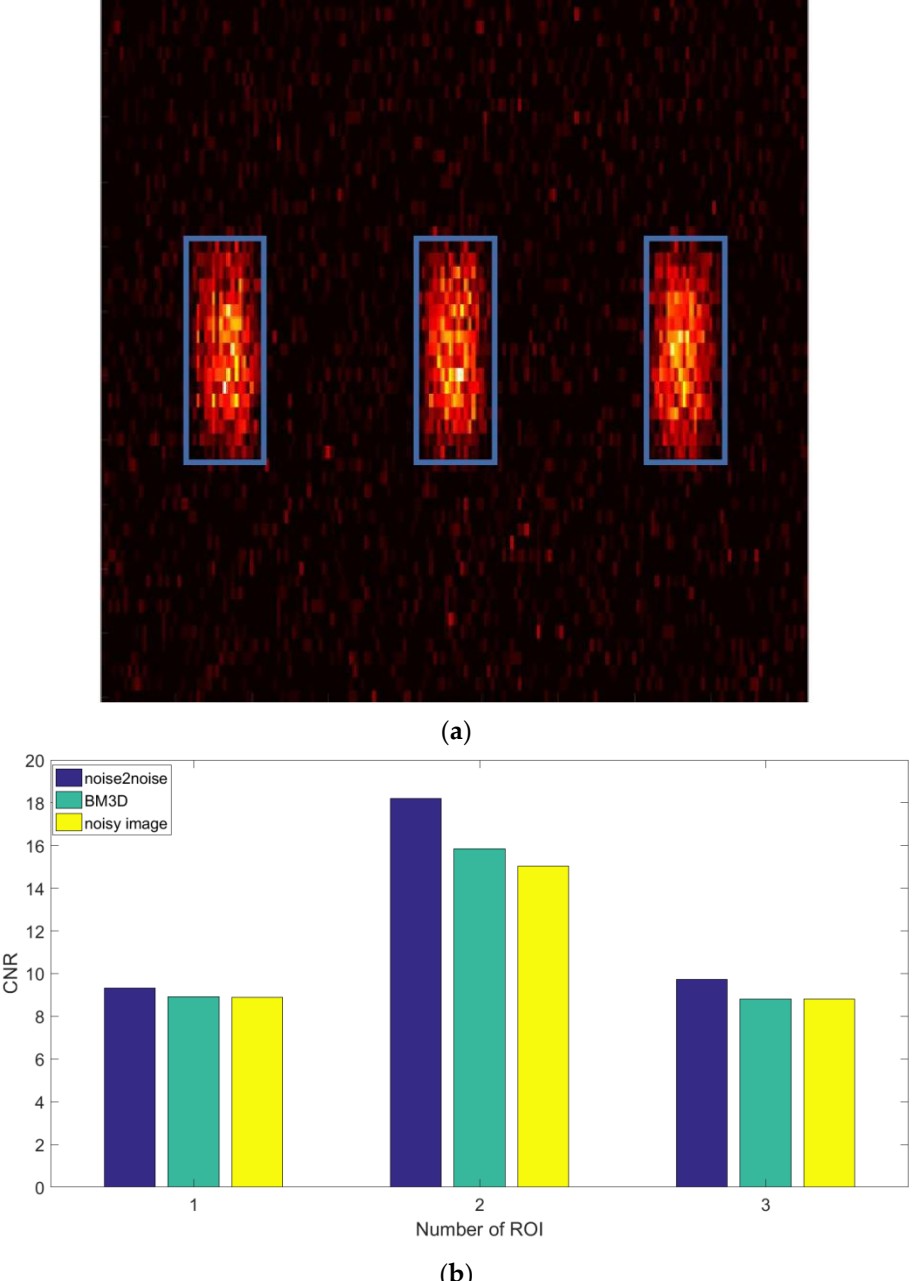

(**a**)

(**b**)

**Figure 7.** CNR of test sets. (**a**) ROI distribution, from left to right are ROI1, ROI2, ROI3; (**b**) comparison of CNR of noisy image and denoised image.

## 4. Discussion

Because the data set is composed of randomly selected photons from different incident events scanning the Gd phantom, the model may have limitations for its single noise level and phantom shape. Therefore, we changed the shape of the phantom, increased the number of ROIs, and changed the material of the phantom. The element and concentration of the phantom are used to examine the generalization ability of the model. Because the Gd element is suitable for imaging in the high concentration range, the imaging quality is the best. The Bi element is suitable for imaging in the low concentration range [15]. In order to make the model more practical, we changed the element type and concentration and migrated the Gd training model to the Bi phantom. The phantom is shown in Figure 8a. A total of 400 pictures are obtained. The Gd phantom model is transferred, 320 pictures are used for training, and 80 pictures are used as the test set.

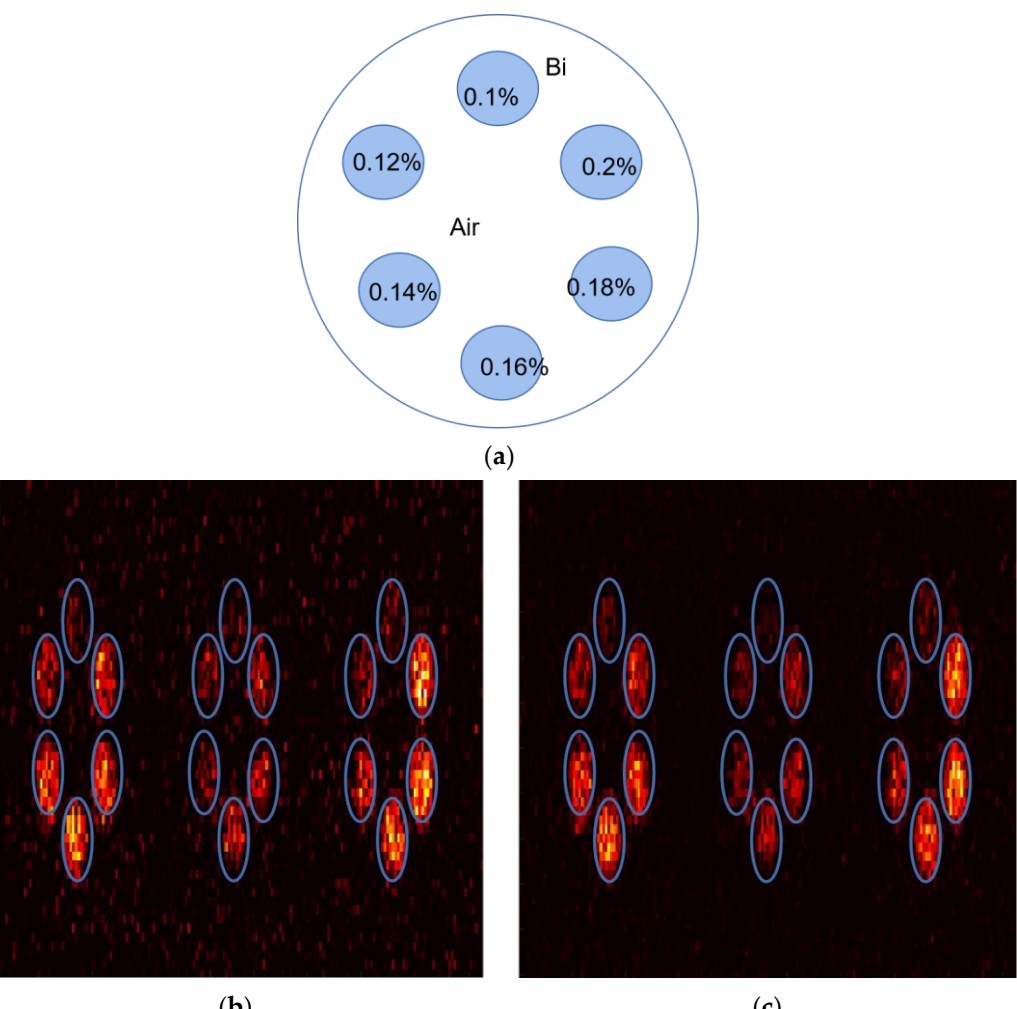

**Figure 8.** Images of low concentration Bi phantom. (**a**) phantom diagram; (**b**) noisy image; (**c**) denoised image.

The noise reduction effect is shown in Figure 8b,c. Due to the low concentration, the image quality is poor. Because the number of detector units in the row direction and column direction are different when the detector is set in Geant4, the ROI in the image is displayed as an ellipse. As can be seen from the figure, when the concentration of the contrast agent is less than 0.1%, the noise and fluorescence signal in the ROI area cannot be clearly distinguished. The Compton background noise is effectively removed after denoising, and the ROI region is more uniform. The image quality is significantly enhanced at the concentration of 0.16%, 0.18%, and 0.2%.

The 175th column of the noisy image and the denoised image are displayed, marked by a blue line in Figure 9a. The pixel value of the spike noise is close to the background noise, and the noise has been removed to a great extent. We calculate the CNR value of each ROI in the test set and take the average value. As shown in Figure 9c, the CNR value of each ROI has increased, which proves that this network is fit for different noise levels and different shape phantoms.

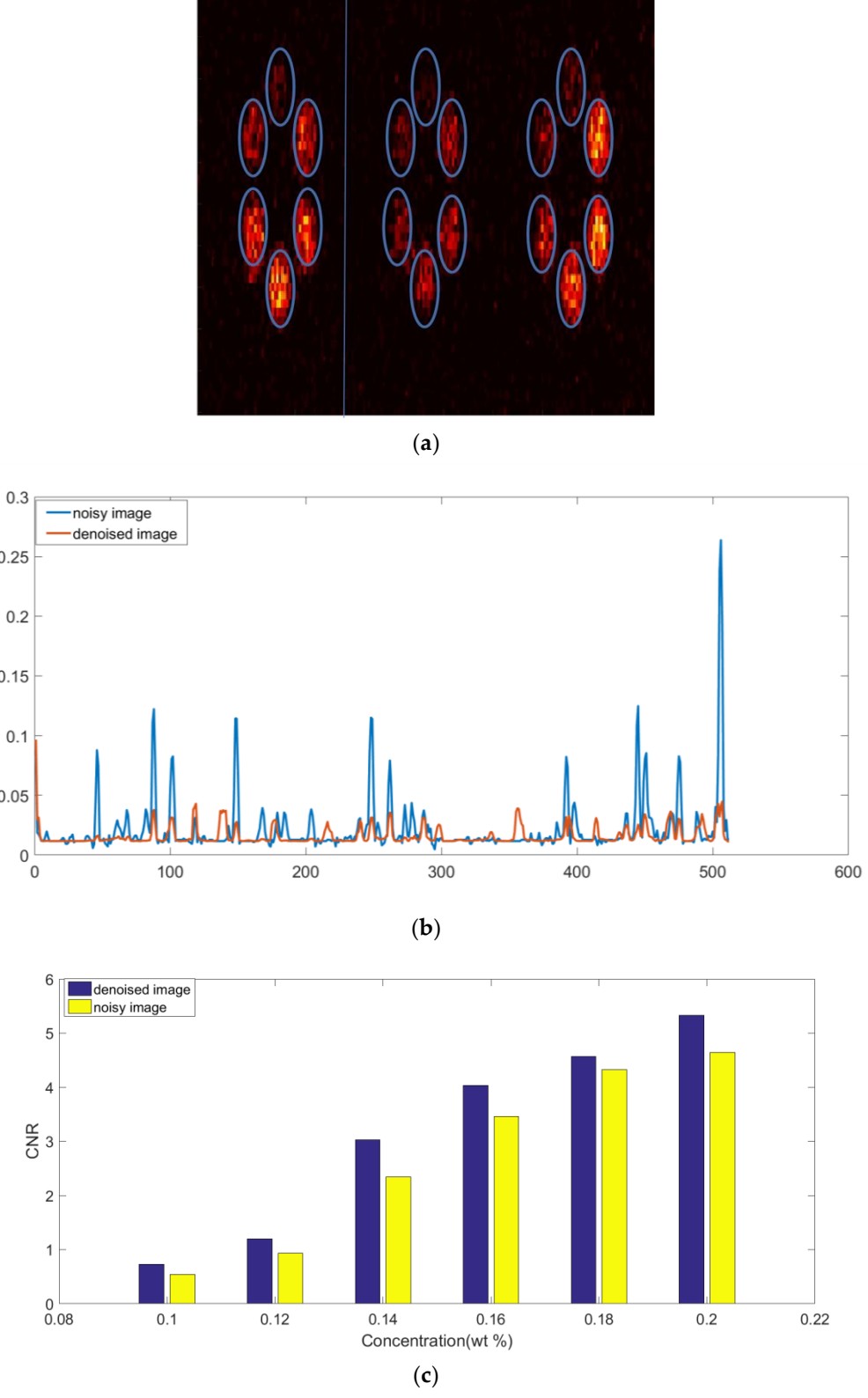

**Figure 9.** Denoising effect of Bi phantom. (**a**) denoised image (blue line is the 175th row); (**b**) pixel profile of 175th row; (**c**) CNR comparison of the denoised images and noisy images.

It seems that the denoising effect of our algorithm in this paper has not been significantly improved, which should be due to two aspects: first, the use of fast pinhole imaging

leads to the significant reduction of the signal-to-noise ratio of the collected fluorescence signal compared with the traditional fan beam or cone beam imaging, thus, the algorithm needs to be further optimized; second, the training data generated by simulation in this paper is still too small to obtain enough features to eliminate the background noise. At the same time, the type of phantom can be further increased to make it closer to the real sample, so as to improve the generalization of the training model.

## 5. Conclusions

In this paper, a noise2noise model based on a deep learning framework is used to denoise XFCT images with different noise levels. The model is based on pinhole XFCT imaging modality and does not need a clean image as the ground-truth image. The noise distribution characteristics are obtained through the learning of noisy images, so as to suppress Compton background. The dataset generated with a high concentration Gd phantom is simulated by GEANT4, and the effectiveness of the model in removing Compton background noise from the XFCT image is verified in terms of pixel profile and CNR. In order to verify the generalization degree of the model, we changed the shape of the phantom, the type, and the concentration of nanoparticles. The denoising effect of low concentration Bi is still good, which shows the feasibility of XFCT denoising using the noise2noise model.

**Author Contributions:** Conceptualization, P.F., and Y.L. (Yan Luo); methodology, P.H. (Pan Huang); software, Y.L. (Yan Luo); validation, Y.L. (Yan Luo), R.Z., and P.H. (Peng He); formal analysis, P.H. (Peng He), and B.T.; investigation, Y.L. (Yan Luo), and Y.L. (Yonghui Li); resources, P.F.; data curation, R.Z.; writing—original draft preparation, Y.L. (Yan Luo); writing—review and editing, X.Z. All authors have read and agreed to the published version of the manuscript.

**Funding:** This work was partially supported by the National Key R&D Program of China (2019YFC0605203), the graduate research and innovation foundation of Chongqing (Grant No. CYB21059 and No. CYS21059), and the Chongqing Basic Research and Frontier Exploration Project (cstc2020jcyj-msxmX0553).

**Institutional Review Board Statement:** Not applicable.

**Informed Consent Statement:** Not applicable.

**Data Availability Statement:** Data sharing not applicable.

**Conflicts of Interest:** The authors declare no conflict of interest.

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
