# Peer review of "Reduction of Compton Background Noise for X-ray Fluorescence Computed Tomography with Deep Learning"

_photonics, doi:10.3390/photonics9020108_

Round 1

Reviewer 1 Report

This manuscript presents a noise2noise denoising algorithm based on Unet deep learning network, aiming at suppressing the Compton background noise of x-ray fluorescence CT. This kind of denoising method is a novel try in XFCT, which mostly makes subtraction operation to get clean XFCT image. The result is interesting and impressive. But there are still some questions the authors should address.

  1. In Experimental setup section, the authors should add more information about the network, at least including hyper-parameters, iteration number and so on.
  2. In the Discussion section, for the low concentration Phantom 2, the authors give the pixel profile of the 175th column in the image. I suggest that the pixel profile of a certain row should also be given in order to observe the denoising effect of ROI area.
  3. In Figure 6 and Figure 9 (b), what is the abscissa in the figure? What is the ordinate?
  4. In Figure 7 (b), the left title is CNR and the top title is CNR, too. Is there any difference between the two CNR?
  5. There are also misspelling words, the authors should double check whole manuscript.
  6. I recommend that authors rewrite section 2.1 and redraw figures 1 and 2, which have been used in other articles of the authors.
  7. I suggest the authors should polish whole manuscript with the help of English native.

Author Response

We would like to thank you for reviewing our manuscript and propose the construction suggestions. We have improved our manuscript by following your comments. Please refer to our detailed responses (in red and italic) below which address all of the reviews’ comments (in italic). All revised parts in manuscript are highlighted with blue color. We trust that the revised manuscript address your critiques sufficiently.

Reviewer 2 Report

In this manuscript, the authors P. Feng et al. propose an algorithm for denoising raw projection images for XFCT. Compton background is a significant issue in XFCT, so the algorithm presented in the manuscript is conceptually interesting

The language in the manuscript should be improved. Below are some specific comments that should also be adressed:

Abstract: It is not clear from the text that the data generated in this study is from a Geant4 simulation. In my first read through I assumed the authors had generated data from an experimental imaging system. I suggest adding “Geant4-simulated data” or similar to the abstract to make this more clear.

Line 24: The authors write: “(XFCT) is a novel method to detect early stage cancer”, although none of the references cited here ([1-3]) actually demonstrate XFCT for detecting cancer. References [1-3] only demonstrate imaging on phantoms, which is misleading. Instead, I suggest citing the following papers instead where the cancer imaging application has been demonstrated in mice:

  • Zhang, Siyuan, et al. "Quantitative imaging of Gd nanoparticles
    in mice using benchtop cone-beam X-ray fluorescence computed
    tomography system." International journal of molecular sciences
    20.9 (2019): 2315.

  • Larsson, Jakob C., et al. "High-spatial-resolution x-ray fluorescence tomography with spectrally matched nanoparticles." Physics in Medicine & Biology16 (2018): 164001.
  • Manohar, Nivedh, et al. "Quantitative imaging of gold nanoparticle distribution in a tumor-bearing mouse using benchtop x-ray fluorescence computed tomography." Scientific reports1 (2016): 1-10.

Lines 32-41: The authors discuss the problem with Compton background which is indeed one of the main limiting factors of XFCT. However, experimental methods to reduce the Compton background are not mentioned at all. For instance, state-of-the-art pencil-beam based XFCT systems (see references below) using multi-layer mirrors have significantly lower Compton background. This should be discussed by the authors, for instance by referring to the latest development in pencil-beam XFCT:

  • Shaker, Kian, et al. "Longitudinal In-Vivo X-Ray Fluorescence Computed Tomography with Molybdenum Nanoparticles." IEEE Transactions on Medical Imaging12 (2020): 3910-3919.

Line 72: Can the authors write “GNPs”, although gold is not used as a contrast agent in this manuscript (but only Gd and Bi). I assume the theoretical equations are general for all elements? I suggest changing “GNPs” to “the contrast agent (e.g., here Bi and Gd)” or similar.

Line 77: Has the “fast multi-pinhole collimated XFCT system” been published before? If yes, provide the relevant reference here.

Line 129: “Experimental setup” is misleading, as the authors merely describe the network architecture and not an experimental setup. Better to label this subsection “Network architecture” or similar.

Results:

What was the rationale behind comparing the proposed algorithm with the BM3D algorithm? Is this the state-of-the-art denoising algorithm currently available? Please elaborate.

The increase in CNR (Fig. 7 and 9) is not so substantial compared to the noisy images. How do the authors interpret this? In fact, it is possible to improve CNR by just subtracting a constant value from the whole image (e.g., subtracting both signal and background). Any comments on this?

Author Response

(The authors gave the same response as above.)
